# Using Ambulatory Care Sensitive Conditions to Assess Primary Health Care Performance during Disasters: A Systematic Review

**DOI:** 10.3390/ijerph19159193

**Published:** 2022-07-27

**Authors:** Alessandro Lamberti-Castronuovo, Martina Valente, Chiara Aleni, Ives Hubloue, Luca Ragazzoni, Francesco Barone-Adesi

**Affiliations:** 1CRIMEDIM—Center for Research and Training in Disaster Medicine, Humanitarian Aid and Global Health, Università del Piemonte Orientale, 28100 Novara, Italy; martina.valente@uniupo.it (M.V.); luca.ragazzoni@med.uniupo.it (L.R.); francesco.baroneadesi@uniupo.it (F.B.-A.); 2Department for Sustainable Development and Ecological Transition, Università del Piemonte Orientale, 13100 Vercelli, Italy; chiara.aleni@uniupo.it; 3Research Group on Emergency and Disaster Medicine, Vrije Universiteit Brussel, 1050 Brussels, Belgium; ives.hubloue@vub.be; 4Department of Translational Medicine, Università del Piemonte Orientale, 28100 Novara, Italy

**Keywords:** ambulatory care sensitive conditions, avoidable hospitalizations, disasters, primary health care, vulnerable populations, H-EDRM

## Abstract

Ambulatory care sensitive conditions (ACSCs) are health conditions for which appropriate primary care intervention could prevent hospital admission. ACSC hospitalization rates are a well-established parameter for assessing the performance of primary health care (PHC). Although this indicator has been extensively used to monitor the performance of PHC systems in peacetime, its consideration during disasters has been neglected. The World Health Organization (WHO) has acknowledged the importance of PHC in guaranteeing continuity of care during and after a disaster for avoiding negative health outcomes. We conducted a systematic review to evaluate the extent and nature of research activity on the use of ACSCs during disasters, with an eye toward finding innovative ways to assess the level of PHC function at times of crisis. Online databases were searched to identify papers. A final list of nine publications was retrieved. The analysis of the reviewed articles confirmed that ACSCs can serve as a useful indicator of PHC performance during disasters, with several caveats that must be considered. The reviewed articles cover several disaster scenarios and a wide variety of methodologies showing the connection between ACSCs and health system performance. The strengths and weaknesses of using different methodologies are explored and recommendations are given for using ACSCs to assess PHC performance during disasters.

## 1. Introduction

In recent years, a new common and comprehensive approach to disaster management (DM) has emerged, which focuses on prevention and preparedness strategies. This new approach promotes coordination across an entire health system, as well as between the health system and other sectors [1]. In 2019, the World Health Organization (WHO) published the Health Emergency and Disaster Risk Management (H-EDRM) framework, detailing how this comprehensive approach can be implemented by stakeholders in all sectors working to reduce health risks and consequences of disasters [2]. In particular, the H-EDRM framework’s “whole-of-health-system” approach to DM stresses the need for each level of a health care system to support disaster preparedness, including the primary health care (PHC) system.

The PHC system can take on a variety of crucial roles during a disaster [3]. It can be the first point of entry to health care, addressing the health needs and priorities of vulnerable and marginalized groups who usually bear the worst consequences of a disaster. It can also prevent the overburdening of hospital and emergency services by maintaining continuity of care for those with minor injuries and non-emergent conditions. Lastly, it can reduce the severity of the surge in demand for chronic care needs in the extended period following the acute response [4].

Hospitalization rates for ambulatory care sensitive conditions (ACSCs) have proven to be a reliable proxy for measuring the quality and accessibility of PHC systems [5,6,7,8]. The concept of ACSCs was first introduced in the late 1980s by John Billings of New York University as a means for describing the differences in access and utilization of health services among vulnerable populations [9]. According to Billings’ definition, ACSCs are “health conditions-diagnoses for which timely and effective outpatient care can help reduce the risks of hospitalization by either preventing the onset of an illness or condition, controlling an acute episodic illness or condition, or managing a chronic disease”. Elevated ACSC hospitalization rates suggest that there are problems in access to care, especially for disadvantaged and marginalized populations. Surveillance of hospital admissions for these conditions may thus be used as a population-based indicator both to evaluate and compare the quality and utilization of PHC and to identify subgroups not receiving adequate and timely access to PHC [10,11].

The literature on ACSCs has been growing over the past ten years. Besides methodological publications aimed at defining country-specific lists of ACSCs [12] and those unraveling methodological implications and potential pitfalls deriving from the use of ACSCs [13], the published original studies have primarily aimed at comparing hospitalization rates due to ACSCs across different groups or countries [14,15,16], performing economic evaluations of avoidable hospitalizations [17,18], identifying inequalities in terms of access to ambulatory care [19,20], and examining trends over the years to spot temporal increases and geographic variations [21,22]. Despite the vast body of literature using ACSC hospitalizations as an indicator of PHC performance and functioning, the literature examining ACSC hospitalizations during disasters and public health emergencies is scarce.

Measuring the performance of a PHC system during different phases of a disaster can contribute to building health systems that are more effective and resilient through all stages of DM. Although ACSC indicators are useful for understanding PHC performance, their use in disaster epidemiology has not been thoroughly examined. As of yet, there is no clear indication of how such indicators can be used to advance this field of research. The aim of this review is to understand to what extent ACSC hospitalization rates during disasters have been studied, and how accurately they can be used to assess PHC performance.

## 2. Materials and Methods

### 2.1. Search Strategy and Eligibility Criteria

This review was conducted in January 2022, following the Preferred Reporting Items for Systematic reviews and Meta-Analyses (PRISMA) checklist. The PubMed, Web of Science, and Scopus databases were searched to find relevant publications. The following were the main eligibility criteria for articles in the study: (i) the study measures ACSC hospitalizations during a disaster, either to assess PHC performance or not; (ii) the study indicates the list of ACSC diagnoses that were considered; and (iii) the study is written in English, Italian, Portuguese, French, German, Arabic, or Romanian. Articles were therefore excluded whenever they focused on ED encounters/calls rather than hospitalizations, targeted non-disaster situations, or did not report a list of ACSC diagnoses. In order to obtain a comprehensive overview of the subject, no restrictions were applied regarding the study type or year of publication. Titles, abstracts, and, when applicable, MeSH terms were searched by combining two groups of key words, namely, ACSC- and disaster-related keywords, using Boolean operators AND and OR (Appendix A). The search and screening of titles and abstracts were performed by authors ALC, CA, and MV, against the agreed inclusion and exclusion criteria; disagreements between reviewers were resolved by consensus. Because of the surge in publications on health system performance during the COVID-19 pandemic, a second search was performed in April 2022, to account for potential new publications measuring ACSCs during the pandemic. This second search was performed using the same search query and databases but restricting the search to 2022 publications.

### 2.2. Data Extraction and Synthesis

Data were extracted from the retrieved articles and recorded into an extraction sheet (Appendix A). Extracted information was analyzed to evaluate the extent, scope, and nature of available evidence on the use of ACSC indicators during disasters.

## 3. Results

A total of 1608 records were identified from the database search. Duplicates (*n* = 846) were removed, leaving 762 articles to be judged for relevance. Of those, 751 were excluded, leaving 11 records whose full text was screened. After evaluation, eight studies were included in the first step of the review (Figure 1). The same screening process was then replicated in the context of a second search performed a few months later, which yielded a total of *n* = 1 article. Details on the second screening process are also reported in Figure 1. In total, *n* = 9 articles were included in this review. Among the included studies, five studies [23,24,25,26,27] were conducted in the United States, two [28,29] in Japan, one in Taiwan [30], and one in Canada [31]. The most recent studies were published in 2021–2022 [23,26,27,31], and two more were published in the years 2017 and 2018 [28,29], while the remaining three were published before 2013 [24,25,30]. Five of the included studies [23,26,27,30,31] refer to slow-onset disasters (i.e., epidemics), while the other four [24,25,28,29] refer to sudden-onset disasters (i.e., earthquake, chemical spill). Two studies [25,30] used interrupted time series analysis, two [23,24] were conducted as a pre-/post-test study, three had a cross-sectional design [26,27,28], and two had a retrospective cohort or population-based design [29,31]. All studies were aimed at assessing the impact of disasters on ACSC hospitalization rates, except for Wright et al. (2021), which compared health care use and costs of individuals enrolled in Medicaid before the COVID-19 pandemic vs. those enrolled in Medicaid during the first COVID-19 wave. A breakdown of the full characteristics of the included studies is listed in the (Appendix A).

### 3.1. Target Populations

The populations and source data in the included studies were highly heterogeneous. Two studies [24,25] collected data about hospital and emergency department admissions from US Medicaid beneficiaries (medically vulnerable individuals) and used data from Medicaid paid claims. One study [23] focused on non-COVID-19 patients admitted to a non-intensive care unit at a tertiary hospital, using discharge data from electronic health records. Two studies [28,29] analyzed ACSC patients’ admissions to the hospital; Sasabuchi et al. (2018) collected data from the National Health Insurance of Kumamoto Prefecture and the Late Elders’ Health Insurance, while Sasabuchi et al. (2016) collected data from the Japanese Diagnosis Procedure inpatient database. One study [30] had a population-based sample and relied on inpatient claims of every National Health Insurance beneficiary in Taiwan. Another study [26] collected de-identified insurance claims for adults enrolled in a US-based health maintenance organization network. One study [31] used Canada’s National Ambulatory Care Reporting System to obtain population-based data on ACSC hospitalizations. The last study [27] used North Carolina Medicaid claims data and examined different patterns of Medicaid enrollees.

### 3.2. Types of ACSCs

In total, 18 ACSCs were used in the nine included studies (Appendix A). These can be classified into three groups: (a) chronic ACSCs (i.e., chronic obstructive pulmonary disease (COPD), asthma, congestive heart failure, arterial hypertension (AH), diabetes, angina), (b) acute ACSCs (i.e., dehydration, urinary tract infection, anemia, gastrointestinal bleeding, nutritional deficiencies, seizures, ear–nose–throat infection, cellulitis, pelvic inflammatory disease, dental conditions, gangrene), and (c) vaccine-preventable ACSCs (i.e., influenza, pneumonia). Two studies [28,29] analyzed all three classes independently; three studies [24,25,26] combined the three classes; one study only considered acute and chronic ACSCs [27]; one study [30] analyzed only chronic ACSCs; and two studies [23,31] analyzed chronic ACSCs, making sure to exclude cases of hospital-acquired pneumonia. For a full list of ACSCs used in each study, see Appendix A.

### 3.3. Studies’ Objectives and Main Results

Five studies examined slow-onset disasters, namely, the 2003 SARS epidemic in Taiwan and the COVID-19 pandemic. Leuchter et al. (2021) compared a 6-month period in 2019 with the same 6-month period in 2020 and found a decrease in hospitalization rates for ACSCs during the COVID-19 pandemic. Huang et al. (2009) performed a population-based interrupted time series analysis to compare actual vs. predicted ACSC hospitalizations during the SARS outbreak, finding lower-than-predicted hospitalization rates for all the selected ACSCs during the SARS outbreak, but increased hospitalization rates for diabetes and hypertension following the period of the outbreak. Becker et al. (2022) compared hospitalization rates before the COVID-19 pandemic (March 2019–February 2020) and during the pandemic period (March 2020–February 2021), finding an overall reduction in all hospitalizations (both non-ACSC and ACSC), with a greater reduction for respiratory-related ACSC hospitalizations. Wright et al. (2021) explored health care use and costs among different types of Medicaid enrollees before and during the COVID-19 pandemic, finding that new Medicaid enrollees during COVID-19 were less likely to be hospitalized for ACSCs than those who enrolled before the COVID-19 pandemic. Rennert-May et al. (2021) compared the most frequent ACSC diagnoses for hospital admissions before and after the implementation of COVID-19 public health measures to determine the impact of COVID-19 on hospital admissions and found that, although hospital admissions did not vary significantly, there was a significant reduction in admissions for chronic respiratory conditions.

The sudden-onset disasters considered in the retrieved studies were the 2005 chlorine spill in Graniteville, South Carolina, the 2011 Great East Japan Earthquake, and the 2016 Kumamoto earthquakes in Japan. Regarding the chlorine spill, Runkle et al. (2012) applied an interrupted time series analysis of 36 months in the pre-disaster phase and 24 months in the post-disaster phase, finding a decrease in ACSC hospital and emergency department visits. However, a second analysis of the same population, published by the same authors and considering a wider pool of variables, subsequently found a significant increase in post-disaster hospital and emergency department visits, as well as a significant decline in emergency department discharges. Regarding the 2011 Great East Japan Earthquake, Sasabuchi et al. (2016) compared pre-disaster (July 2010–February 2011) with post-disaster ACSC hospitalization rates (July 2012–February 2013) and found a significant increase in acute ACSC admissions, with no increase in admissions for preventable or chronic ACSCs. A retrospective cohort study performed by Sasabuchi et al. (2018) examined ACSC admissions between 15 March and 16 May of each year from 2013 to 2016 to explore the impact of the 2016 Kumamoto earthquakes and identified an increase in admissions within seven days after the earthquakes for vaccine-preventable, acute, and chronic ACSCs, with no significant change 30 days after the earthquakes.

### 3.4. Vulnerable Populations

Although ACSC hospitalization rates can be used to infer information about demographic and socio-economic factors hampering access to PHC, only Leuchter et al. (2021) and Wright et al. (2021) provided information on disadvantaged groups, namely, those more likely to be hospitalized for ACSCs: African Americans and rural populations.

## 4. Discussion

The aim of this study was to investigate the extent to which ACSCs have been used in the disaster-related literature, and then to shed light on the potential use of such indicators for the assessment of PHC performance during disasters. The fact that only nine articles matched the inclusion criteria shows that ACSC indicators have rarely been used in disaster research. However, as four out of nine articles were published in the last two years following the global spread of COVID-19, scholarly interest in the topic may be increasing. Scholars should approach the use of the ACSC methodology in disaster research with caution. Following disasters, many factors influence hospitalization for ACSCs, such as disease prevalence, patient health-seeking behavior, the degree of damage to health facilities, and the pre-existing health system organization. Consequently, changes in hospitalization rates for ACSCs and the types of conclusions that can be drawn are highly context-dependent.

Overall, hospitalization rates for ACSCs during disasters were measured to achieve the following main goals: (1) to assess the long-term impact of disasters on public health; (2) to evaluate the performance of the PHC system during disasters; and (3) to identify vulnerable categories of health seekers, namely, those more likely to be hospitalized for ACSCs. Regarding the first goal, studies have shown that patterns of avoidable hospitalizations for ACSCs persisted in the long term after the disaster, explaining the so-called “secondary surge” in health needs that follows in the wake of a disaster. In the works of Runkle et al. (2012, 2013), for example, a sudden-onset disaster caused a secondary surge in PHC needs in the weeks and months after the disaster. Regarding the second goal, studies identified impaired access to PHC services as a possible explanation for the changes in hospitalization rates and, thus, used hospitalization rates for ACSCs as a proxy for PHC functioning. Studies on slow-onset disasters [23,26,27,30,31] reported an overall decrease in ACSC hospitalizations, suggesting better management of health needs at the PHC level during sudden-onset rather than slow-onset disasters. Regarding the third goal, studies examined hospitalizations for ACSCs to identify vulnerable groups, namely, those more likely to be hospitalized for ACSCs. Leuchter [23] and Huang [30] showed that, despite an overall reduction in ACSC hospitalization rates, African Americans and diabetic patients experienced higher rates of hospitalization for chronic ACSCs compared to other populations during the COVID-19 and SARS outbreaks.

Caution should be used when considering hospitalizations for ACSCs as a proxy for PHC functioning during disasters. In fact, many complicating factors influence the relationship between PHC and ACSC hospitalizations following disasters, both at the patient and health system levels. Therefore, ACSC data must be interpreted alongside other information if they are to yield accurate results about PHC performance [6].

Patients’ propensity to seek care in the aftermath of a disaster must be considered. In the event of a disaster, patients are more likely to avoid hospital contact [32]. Those with a chronic disease may only seek care later during flare-ups, which makes it more likely for them to be hospitalized at a later stage [33]. This may decrease the rates of potentially avoidable hospitalizations observed during disasters and alter the relationship between ACSCs and PHC performance. This tendency was observed at the beginning of the COVID-19 pandemic and during the SARS outbreak in 2009 [34,35,36]. The measurement of ACSC hospitalizations should include longer time periods for additional time series analyses in order to uncover potentially unrecognized underlying trends that may explain the results, and to appropriately factor the propensity for hospitalization into the analysis.

Differences in admission/discharge hospital policies should also be considered. During disasters, there may be changes to such policies or changes in clinicians’ conduct that need to be considered since they can alter the interpretation of the ACSC data. As an example, during the COVID-19 pandemic, hospital beds were frequently converted to COVID-19 beds for infected patients, lowering the number of beds available for non-COVID cases [37]. Moreover, physicians might have hesitated to admit or discharge patients amid the pandemic because of a fear of transmission, a lack of possibilities of isolation, or due to a patient’s lack of social support. Moreover, patients may be re-hospitalized in nearby hospitals, so re-admissions should also be considered, and hospitals’ health information systems should be used to capture this phenomenon and avoid a misinterpretation of data.

The degree of damage to health facilities is another disaster-specific factor that needs to be considered. Information about the degree of damage to hospital facilities (i.e., number of available beds after the disaster vs. standard in-hospital bed capacity) and the degree of disruption to the service of health care facilities needs to be considered to accurately interpret the rates of hospitalization for ACSCs.

The organization of a health system is also relevant when interpreting ACSC data during disasters. When the health system of a country grants access to all patients, regardless of legal entitlements or health conditions, treatment can be administered before health problems become severe, thus reducing rates for hospitalizations [38]. In contrast to what can be generally observed in countries such as the US, where disasters caused by natural hazards usually result in higher rates of hospitalizations for chronic ACSCs [39,40], Japan saw no changes in admissions for chronic ACSCs after the earthquakes in the studies of Sasabuchi et al. (2016, 2018). One possible explanation for this discrepancy may be the fact that, in Japan, PHC centers had absorbed the unmet primary care needs (i.e., chronic disease management) that were present after the earthquake because Japan’s health system is based on universal health coverage, as opposed to the insurance-based systems in the US.

To the best of our knowledge, this is the first systematic review addressing the use of ACSCs in disaster research. However, the scant number of studies that met the inclusion criteria demonstrates that the topic is still relatively unexplored. This study was limited by the fact that only publications from languages used by the co-authors and co-researchers were included. However, no potentially relevant article written in any other language (analyzed through the abstracts available in English) could be retrieved. Since this field of inquiry is still in its infancy, each of the reviewed articles used distinct methods and approaches. Therefore, it is still very difficult to generalize about the use of ACSCs in disaster research.

## 5. Conclusions

In agreement with the H-EDRM framework, it is important to consider the functioning of PHC systems during disasters to plan evidence-based interventions aimed at strengthening PHC disaster preparedness and DM at the PHC level. In the absence of direct methods to assess PHC performance during disasters, the assessment of avoidable hospitalizations can be a reliable strategy to monitor the performance of PHC in times of crisis. Nevertheless, caution should be used when interpreting changes in disaster-related ACSC hospitalization rates as a proxy for PHC performance, as many factors need to be considered to properly interpret ACSC patterns and increasing/decreasing trends.

## Figures and Tables

**Figure 1 ijerph-19-09193-f001:**
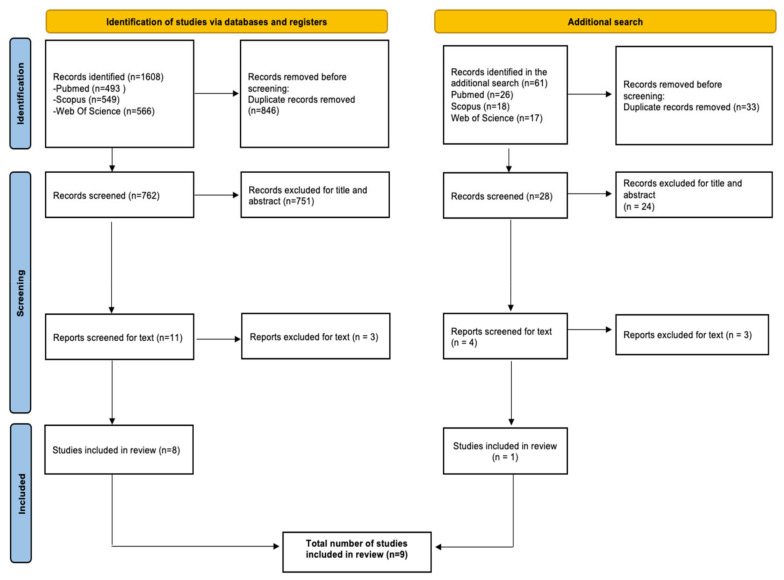
PRISMA checklist for the identification of studies.

## Data Availability

Not applicable.

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
