# Peer review of "Using Ambulatory Care Sensitive Conditions to Assess Primary Health Care Performance during Disasters: A Systematic Review"

_ijerph, 2022, doi:10.3390/ijerph19159193_

Round 1
Reviewer 1 Report
Report on “Using Ambulatory Care Sensitive Conditions to Assess Primary Health Care Performance During Disasters: A Systematic Review” by Alessandro Lamberti-Castronuovo et al.
The manuscript reviewed the literature about research activity on the use of ACSCs during disasters. I think the manuscript is well written and quite clear. The study follows the Preferred Reporting Items for Systematic reviews and Meta-Analyses Checklist PRISMA. The methodology is detailed, results are clear, and the discussion introduce the main findings of the review in the field.
Unfortunately, the manuscript cannot be evaluated, since tables 1 and 2 are missing, and until the tables are presented, a judgment of acceptance cannot be issued.
As a minor concern, the concept of ACSCs is introduced in a unique paragraph in the introduction section. In my opinion, this is not enough for a wide audience such as the readers of this journal. It should be introduced in a more detail way with references to its use.
Author Response
We thank Reviewer 1 very much for taking the time to review our manuscript and apologize for forgetting to attach the tables. We have now attached the two tables as Appendix C and D. To address the minor concern expressed by Reviewer 1, which suggests a more thorough explanation of ACSCs in the introduction, we have reviewed additional literature and reported some more information to better introduce the concept of ACSCs in the introduction of our review.
Reviewer 2 Report
The work entitled “Using Ambulatory Care Sensitive Conditions to Assess Primary Health Care Performance During Disasters: A Systematic Review” by Lamberti-Castronuovo et al. is a review of the literature on the relationship between Ambulatory Care Sensitive Condition (ACSC), a well-established parameter for assessing the performance of Primary Health Care (PHC), for which appropriate intervention in primary care could prevent hospitalization. The authors tried to present a perspective on the existing literature on these issues and the relationships between them. Despite being widely used to monitor the performance of the performance of the PHC system in peacetime, its consideration during disasters has been neglected, so the authors conducted a systematic review to assess the functioning of the PHC during disasters, but only nine studies met the criteria for inclusion in the study, which is interesting.
I think that the manuscript can be accepted after reviewing the presentation of the results, better summarizing the discussions, highlighting new perspectives in Conclusions, and some formal corrections / typos.
I recommend to better clarify the main results of the study through a schematic figure / schematic view in order to summarize all the information obtained.
Regarding the References, I consider that all should be better reformatted in the spirit of the journal, with Digital Object Identifier (DOI) included, or to provide a permanent web address (URL), to easily locate all the documents.
The self-citation from point no. 3: "Lamberti-Castronuovo A, Valente M, Barone-Adesi F, Hubloue I, Ragazzoni L. Primary Care Disaster Preparedness: A Review of the Literature and the Proposal of a New Framework (pre-publication stage)", being in a pre-publication stage, must be deleted.
Finally, I recommend that the paper be reviewed by a native English speaker to correct all embarrassment of expression.
Author Response
We want to deeply thank Reviewer 2 for the useful feedback. We apologize for forgetting to attach the tables to the main file. We have now attached the two tables to the document as Appendix C and D and hope this will facilitate the understanding of the results and address the first comment regarding the need to summarize the results via a “schematic figure/schematic view”. With regard to the references, these have been adapted to comply with the journal’s standards and the articles’ DOI has been added for all the cited publications. Self-citation nr. 3 has been deleted and replaced with the work of Redwood-Campbell et al., 2011. Last, the article underwent an additional revision by a native English speaker.
Reviewer 3 Report
Please see word with comments and suggestions

Author Response
We thank Reviewer 3 for taking the time to review our manuscript and apologize for forgetting to attach the tables. We have now attached the two tables as Appendix C and D. We have revised the eligibility criteria. With regard to the inclusion criteria about the language, we are aware that the exclusion of some languages might have been a source of bias in our study. The choice of languages was made according to the languages spoken by the co-authors and researchers. This limitation has now been acknowledged in the discussion. With regard to the PRISMA diagram, this is now mentioned in the results section.
Round 2
Reviewer 1 Report
All my concerns have been addressed by the authors.